# Global Regulator PhoP is Necessary for Motility, Biofilm Formation, Exoenzyme Production, and Virulence of *Xanthomonas citri* Subsp. *citri* on Citrus Plants

**DOI:** 10.3390/genes10050340

**Published:** 2019-05-06

**Authors:** Chudan Wei, Tian Ding, Changqing Chang, Chengpeng Yu, Xingwei Li, Qiongguang Liu

**Affiliations:** 1College of Agriculture, South China Agricultural University, Guangzhou 510642, China; weichudan3@163.com (C.W.); changcq@scau.edu.cn (C.C.); ycp9393@163.com (C.Y.); xingwli@163.com (X.L.); 2Guangzhou Airport Entry-Exit Inspection and Quarantine Bureau, Guangzhou 510800, China; czdtian@163.com; 3State Key Laboratory of Conservation and Utilization of Subtropical Agro-bioresources, Guangdong Province Key Laboratory of Microbial Signals and Disease Control, Guangzhou 510642, China

**Keywords:** *Xanthomonas citri* subsp. *citri*, PhoP, RNA-Seq, motility, biofilm, exoenzyme, virulence regulation

## Abstract

Citrus canker caused by *Xanthomonas citri* subsp. *citri* is one of the most important bacterial diseases of citrus, impacting both plant growth and fruit quality. Identifying and elucidating the roles of genes associated with pathogenesis has aided our understanding of the molecular basis of citrus-bacteria interactions. However, the complex virulence mechanisms of *X. citri* subsp. *citri* are still not well understood. In this study, we characterized the role of PhoP in *X. citri* subsp. *citri* using a *phoP* deletion mutant, Δ*phoP*. Compared with wild-type strain XHG3, Δ*phoP* showed reduced motility, biofilm formation, as well as decreased production of cellulase, amylase, and polygalacturonase. In addition, the virulence of Δ*phoP* on citrus leaves was significantly decreased. To further understand the virulence mechanisms of *X. citri* subsp. *citri*, high-throughput RNA sequencing technology (RNA-Seq) was used to compare the transcriptomes of the wild-type and mutant strains. Analysis revealed 1017 differentially-expressed genes (DEGs), of which 614 were up-regulated and 403 were down-regulated in Δ*phoP*. Gene ontology functional enrichment and Kyoto Encyclopedia of Genes and Genomes pathway analyses suggested that the DEGs were enriched in flagellar assembly, two-component systems, histidine metabolism, bacterial chemotaxis, ABC transporters, and bacterial secretion systems. Our results showed that PhoP activates the expression of a large set of virulence genes, including 22 type III secretion system genes and 15 type III secretion system effector genes, as well as several genes involved in chemotaxis, and flagellar and histidine biosynthesis. Two-step reverse-transcription polymerase chain reaction analysis targeting 17 genes was used to validate the RNA-seq data, and confirmed that the expression of all 17 genes, except for that of *virB1*, decreased significantly. Our results suggest that PhoP interacts with a global signaling network to co-ordinate the expression of multiple virulence factors involved in modification and adaption to the host environment during infection.

## 1. Introduction

Citrus crop canker, caused by *Xanthomonas citri* subsp. *citri* and generally associated with a characteristic embossment of necrotic lesions on infected leaves, stems, and fruit [1,2], is considered one of the most serious crop diseases worldwide [3]. Citrus yields and quality are greatly impaired by citrus canker because of defoliation, blemished fruit, and in serious cases, premature fruit drop [4]. *X. citri* subsp. *citri* invasion of the citrus host occurs directly through natural openings, including stomata, or wounds, with the bacteria then acquiring nutrients from host cells and proliferating in the apoplast [4]. *X. citri* subsp. *citri* is an important model *Xanthomonas* pathogen and is used in studies investigating plant-microbe interactions and virulence mechanisms. The pathogen has evolved several strategies, including a type III secretion system (T3SS), extracellular enzymes, and polysaccharides, to adapt to and successfully establish an *in planta* niche, conquering plant defenses and creating a favorable environment for growth [5,6]. Previous studies have characterized the major pathogenicity and virulence genes responsible for secretion systems such as the T3SS components and effector molecules, as well as bacterial adhesins, extracellular enzymes, toxins, surface structural elements, and *rpf* (regulation of pathogenicity factors)-encoded cell-cell signaling proteins [7,8,9,10,11,12]. PthA, an effector of the T3SS, is a critical factor for citrus canker symptoms [13], while GalU, a UTP-glucose-1-phosphate uridylyltransferase, contributes to the growth of *X. citri* subsp. *citri* in intercellular spaces and is involved in the synthesis of extracellular polysaccharide (EPS) and lipopolysaccharide (LPS), as well as in biofilm formation [14]. Genome sequencing of *X. citri* subsp. *citri* has greatly increased our understanding of *X. citri* subsp. *citri*-citrus plant interaction [7,15].

Bacterial two-component regulatory systems (TCS) regulate a wide gamut of biological processes in response to fluctuating environmental stimuli. However, only a few *X. citri* subsp. *citri* TCS have been investigated for their contributions to virulence. For example, the HrpG/HrpX TCS of *X. citri* subsp. *citri* interacts with a global signaling network to co-ordinate the expression of multiple virulence factors required for the modification of and adaption to the host environment during infection [8]. The ColR/ColS system is also critical for *X. citri* subsp. *citri* virulence, contributing to growth *in planta*, biofilm formation, LPS production, catalase activity, and resistance to environmental stress [6]. PhoQ/PhoP, which modulates adaptive responses to changes in levels of divalent cations, including magnesium, in the environment [16,17], is an evolutionarily active system involved in bacterial adaptation to various ecological niches [18]. In animal-pathogenic bacterium *Salmonella enterica* subsp. *enterica* serovar Typhimurium, antimicrobial peptides serve as direct signals for the activation of PhoQ [19], while regulatory protein PhoP controls susceptibility to the host inflammatory response in *Shigella flexneri* [20]. PhoP also controls virulence in *Yersinia pestis*, the etiological agent of plague, with a *Y. pestis phoP* mutant showing decreased survival within macrophages and increased sensitivity to low pH, oxidative killing, and high osmolarity [21].

In plant-pathogenic bacteria, the PhoQ/PhoP system has been implicated in the regulation of several virulence determinants in *Dickeya dadantii* (formerly known as *Erwinia chrysanthemi*), a pectinolytic enterobacterium causing soft rot in several plant species [22], and is required for *hrpG* expression, the stress defense response, cation transportation, and virulence in *Xanthomonas oryzae* pv. *oryzae* AvrXa21 [23,24]. The expression of the set of genes activated by PhoP varies among different bacterial species [25,26]. Comparison of the downstream genes regulated by PhoP in *S. Typhimurium* and in *Escherichia coli* showed that gene expression was very different between the two species [27]. In *X. campestris* pv. *campestris*, PhoP controls the transcription of many essential, structural genes by directly binding to their *cis*-regulatory elements. However, it does not control the same essential genes in *Pseudomonas aeruginosa*, as no PhoP binding sites exist in the promoter regions of the genes [18]. A comparative analysis of the genomes of *P. aeruginosa*, *X. campestris* pv. *campestris* and *X. oryzae* pv. *oryzae* found that only five genes with expression putatively regulated by PhoP were shared among the three species. Although the PhoP amino acid sequences from *X. campestris* pv. *campestris* and *X. oryzae* pv. *oryzae* were identical, approximately 70% of the PhoP-binding sequences differed between the two bacteria, indicating that the effects and regulatory mechanisms of PhoQ/PhoP may differ amongst pathogenic bacteria, even those belonging to the same genus.

Despite the information available on other species, the function of PhoP in *X. citri* subsp. *citri* remains unclear. Therefore, as well as analyzing PhoP-regulated gene expression in *X. citri* subsp. *citri*, we examined the role of PhoP with respect to virulence in citrus to provide new information about the virulence of this important plant pathogen.

## 2. Materials and Methods

### 2.1. Bacterial Strains, Culture Media, and Growth Conditions

The bacterial strains and plasmids used in this study are listed in Table 1. Wild type strain XHG3 was isolated from citrus cultivar of Xinhugang in Guangdong province of China. This strain has a high virulence on citrus plants and is preserved in Guangdong Province Key Laboratory of Microbial Signals and Disease Control. *Xanthomonas citri* subsp. *citri* was routinely grown at 28 °C in YEB medium (10 g/L tryptone, 5 g/L yeast extract, 5g/L NaCl, 0.5 g/L MgSO_4_·7H_2_O, 5g/L sucrose, pH7.0) or NA (5 g/L tryptone, 10 g/L sucrose, 1 g/L yeast extract, 3 g/L beef extract, 15 g/L agar, pH7.0). Transformants bearing the first crossover for the gene knockout were cultured on NA without 10% sucrose medium. Transformants for the second crossover were plated on NA medium. *E. coli* was grown at 37 °C in LB medium (10 g/L Bacto tryptone, 5 g/L yeast extract, and 10 g/L NaCl, pH7.0). NYGA medium (5 g/L tryptone, 3 g/L yeast extract, 20 mL/L glycerol, 15 g/L agar, pH7.0) was used for enzyme assays. Antibiotics were added at the following final concentrations when required: streptomycin sulphate (Sm), 50 μg∙mL^−1^; kanamycin (Km), 50 μg·mL^−1^; Ampicillin (Amp), 100 μg∙mL^−1^. The optical density of the cultures at 600 nm (OD_600_) was measured by using a UNIC 7200 spectrophotometer (UNIC Apparatus Co). All antibiotics were purchased from Sigma-Aldrich.

### 2.2. Biochemical Reagents, Primers and Other Materials

Primer STARTM HS DNA Polymerase, *Taq* DNA Polymerase, T4 DNA Ligase, a Star Prep Plasmid Miniprep Kit, a Universal DNA Purification Kit, a DNA Fragment Purification Kit, PrimeScriptTM RT Master Mix (for Real Time PCR) and SYBR^®^ Premix Ex *Taq*TM II (Tli RNaseH Plus), as well as restriction enzymes *Spe*I, *Bam*HI, *Hin*dIII, *EcoRI*, and D2000 DNA Ladder Marker, were purchased from TaKaRa Bio. A SV Total RNA Isolation System was purchased from Promega. All primer synthesis and sequencing was performed by Invitrogen. Primer sequences are shown in Appendix A. Susceptible cultivars of *Citrus reticulata*, commonly known as Xinhuigan and Orah were maintained in a greenhouse.

### 2.3. Gene Knockout and Complementation of phoP Gene

A *phoP* deletion mutant was generated from wild-type *X. citri* subsp. *citri* strain XHG3 using homologous recombination [28]. All primers used in the generation of the mutants are shown in Appendix A. The left and right homologous arms of *phoP* were amplified by PCR using genomic DNA from XHG3 as a template and primer pairs *phoP*-U-F/*phoP*-U-R and *phoP*-D-F/*phoP*-D-R, respectively. The two amplified fragments, which flank *phoP*, were fused together by PCR using primers *phoP*-U-F/*phoP*-D-R. Following digestion with *Spe*I and *Bam*HI, the fusion fragment was purified and ligated into suicide vector pKNG101 to obtain recombinant plasmid pKNG-Δ*phoP*. pKNG-Δ*phoP* was then transformed into wild-type strain XHG3 by electroporation, and transformants were selected for on NAN agar medium supplemented with Sm (50 μg/mL). Sucrose-sensitive clones were grown in the absence of antibiotics, and double recombination events were selected for on NAS agar plates supplemented with 10% sucrose. The resulting Δ*phoP* mutant was confirmed by PCR and sequencing.

For complementation of the Δ*phoP* mutant, a 684-bp fragment containing the entire gene of *phoP* was amplified from XHG3 genomic DNA using primers *phoP*-F/*phoP*-R. The resulting amplicon was digested with *Bam*HI and *Hin*dIII and then cloned into the corresponding sites of plasmid vector PBBR1MCS4 so that expression of the target gene was under the control of the *lac* promoter present in the vector. The ligated mixture was transfected into *E. coli* DH5α competent cells, and transformants were selected on LB agar supplemented with 100 μg∙mL^−1^ of Amp. Following PCR-based verification using primers MCS4-F/R, the recombinant plasmid was transformed into Δ*phoP* by electroporation to produce the complemented strain, R-*phoP*.

### 2.4. Bacterial Growth Curve

To examine the growth of bacterial strains, cells from overnight cultures of the tested strains were harvested and were inoculated into fresh YEB broth. The strains were then cultured at 30 °C with shaking at 200 rpm. Every 2 h, 2-mL volumes of each culture were collected to measure the optical density at 600 nm (OD_600_). Three replicates were performed for each strain, and bacterial growth curves were plotted based on the average values.

### 2.5. Motility and Biofilm-Formation Assay

Motility assays were performed as described by [29] with modification. Briefly, bacterial strains were cultured in NB medium overnight to an OD_600_ = 0.5. Aliquots (2 μL) of the bacterial suspensions were then inoculated onto the surface of agar plates (10 g tryptone, 5 g NaCl, and 3.0 g agar per liter) and incubated for 72 h at 28 °C, Motility was then assessed by measuring the diameter of the growth circle around the inoculation site. Each assay was repeated three times, and strains were examined in triplicate.

The ability of the strains to form a biofilm was examined as described by [6] with modification. Briefly, a 10-μL aliquot of bacterial cell culture (OD_600_ = 0.5) was inoculated into tubes containing SONG medium (20 g/L of tryptone, 5 g/L of yeast extract, 5 g/L of NaCl, 2.4 g/L of MgSO_4_·7H_2_O, 0.186 g/L of KCl, 50 ml/L of 40% glycerin) and incubated for 72 h at 28 °C. The culture medium was then discarded, and attached bacterial cells were gently washed three times with distilled water, incubated for 20 min at 60 °C, and then stained with 1.5 mL of 0.1% (w/v) crystal violet for 45 min. The unbound crystal violet was discarded, and the wells were washed with water prior to interpretation.

### 2.6. Determination of Extracellular Enzymes Activity

Bacterial protease, amylase and cellulase activities were measured using protease, polygalacturonase (PG) and cellulase test plates, respectively, as described by [30]. Relative levels of exoenzyme production were assessed by radial diffusion assays. Bacterial strains were grown in YEB at 28 °C overnight, and the supernatants were collected by centrifugation. Thirty microliters aliquots of the filter-sterilized supernatant were added to the wells of the exoenzyme test plates for further cultivation at 28 °C for 24 h. Each assay was repeated three times, with each strain examined in triplicate. Enzyme activity was estimated from the diameter of the zones surrounding the culture supernatants of each strain. 

### 2.7. Pathogenicity Assay

Pathogenicity assays were conducted in a quarantine greenhouse facility. Bacterial strains were cultured in YEB for 24 h at 28 °C. The cells were then pelleted by centrifugation and resuspended in sterile water to an OD_600_ = 0.01. The bacterial suspensions were then inoculated onto the tender leaves of cultivars of Xinhuigan and Orah using needleless syringes [31]. Sterile water was used as a control. The inoculated plants were maintained in the greenhouse at 30 °C, and symptoms were observed for three days following inoculation.

### 2.8. High-throughput RNA Sequencing (RNA-seq) and Data Analysis

Wild-type strain XHG3 and the Δ*phoP* mutant were cultured in YEB medium at 28 °C to OD_600_ = 1.0, with two biological repeats performed for each strain. Total bacterial RNA was extracted using the SV Total RNA Isolation System (Promega), with eluted RNA then DNase-treated using a Turbo DNA-free Kit (Promega) as per the manufacturer’s instructions. The quality and quantity of the total RNA was assessed using agarose gel electrophoresis, spectrophotometry, and an Agilent 2100 Bioanalyzer. Transcriptome library construction and sequencing were performed by Novogene (Beijing, China). Clean read data were obtained following sequencing and data filtering, and resulting reads were mapped to the *X. citri* subsp. *citri* strain 306 reference genome. Gene expression, as indicated by the expected number of fragments per kilobase of transcript sequence per million base pairs sequenced (FPKM), was calculated using HTSeq, with difference in expression between strains Δ*phoP* and XHG3 determined using DESeq.

Differentially-expressed genes (DEGs) were designated based on q-values < 0.005 and a minimum absolute |log_2_(Fold Change)| > 1, and were exported as a tabular file (Appendix A). DEGs were then classified and enriched based on gene ontology (GO) database (http://www.geneontology.org/ and Kyoto Encyclopedia of Genes and Genomes (KEGG) pathway analyses.

### 2.9. Quantitative Real-time PCR (qRT-PCR) Analysis

To verify the RNA-seq results and identify the downstream genes regulated by PhoP, the expression of several genes was examined using qRT-PCR analysis. RNA samples were reverse-transcribed using a PrimeScript RT Master Mix (TaKaRa). Gene-specific primers were designed using DNASTAR software (DNASTAR) to generate 100–250 bp products from the *X. citri* subsp. *citri* genome (Appendix A). qRT-PCR analysis was performed using three biological replicates on a Bio-Rad CFX96 Real-Time PCR System using TaKaRa SYBR Premix Ex *Taq* II (Tli RNaseH Plus) as per the manufacturer’s instructions. The 16S rRNA gene was used as an endogenous control to normalize gene expression data. Target genes included *hrpG* and *hrpX* (T3SS regulators), *avrBs2* and *avrXacE1* (avirulence protein), and *egl0028* (cellulase). The relative fold change in gene expression was calculated using the formula 2^-ΔΔCT^ [32]. Fold change values were log_2_ transformed to allow comparison with values generated from RNA-seq analysis.

## 3. Results

**Mutation of *phoP* does not affect the bacterial growth.** To examine the function of PhoP in *X. citri* subsp. *citri*, we successfully constructed *phoP* knockout mutant Δ*phoP*, along with its complementation strain R-*phoP* (Appendix A). To determine whether mutation of *phoP* affects the growth and proliferation of *X. citri* subsp. *citri*, the growth of the wild-type, mutant, and complementation strains was compared in YEB medium. Compared with the wild-type, Δ*phoP* had a slower growth rate between 4 and 16 h post-inoculation, but reached a cell density similar to that of the wild-type by 20 h post-inoculation (Figure 1). At 24 h post-inoculation, there were no obvious differences in the growth of the three strains.

**Mutation of *phoP* reduces motility and biofilm formation in *X. citri* subsp. *citri*.** Motility assays demonstrated that following incubation at 28 °C for 72 h, the Δ*phoP* mutant showed considerably reduced motility compared with wild-type strain XHG3 (Figure 2A), with a 70.6% reduction in colony diameter compared with that of strain XHG3 (Figure 2B). The motility of complemented mutant strain R-*phoP* was comparable with that of the wild-type, suggesting that PhoP positively regulates bacterial motility.

Biofilms are bacterial communities that are attached to a surface so as to provide protection from deleterious conditions [33,34]. Δ*phoP* showed a significant decrease in biofilm formation in glass tubes compared with wide-type strain XHG3 and the complementation strain R-*phoP* (Figure 3), indicating that PhoP regulates biofilm formation in *X. citri* subsp. *citri*.

**Mutation of *phoP* reduces the production of polygalacturonase, amylase and cellulase in *X. citri* subsp. *citri*.** In many plant pathogenic bacteria, extracellular enzymes, including protease, polygalacturonase (PG), cellulase and amylase, are important virulence factors. To investigate the role of PhoP in *X. citri* subsp. *citri* extracellular enzyme production, we compared the levels of the different exoenzymes in Δ*phoP* and wild type strain XHG3. The results showed a 100% reduction in PG activity (Figure 4B,E), a 37.16% reduction in amylase production (Figure 4C,E), and a 19.05% reduction in cellulase activity (Figure 4D,E), in the Δ*phoP* mutant compared with wild-type strain XHG3. The activities of each of the enzymes were restored to wild-type levels in complementation mutant strain R-*phoP*. Interestingly, there was no significant difference in protease production between Δ*phoP* and XHG3 (Figure 4A,E). These results confirm that PhoP is involved in the production of polygalacturonase, amylase and cellulase, in *X. citri* subsp. *citri*, but does not appear to regulate protease production.

**PhoP is required for the virulence of *X. citri* subsp. *citri* on citrus plants.** The tested bacterial suspensions were inoculated onto the young leaves of two citrus cultivars (Xinhuigang and Orah), which were then maintained in the greenhouse at 30 °C. At 3 days post-inoculation, the leaves inoculated with wild-type strain XHG3 showed obvious canker symptoms, while only very mild symptoms were observed on leaves inoculated with Δ*phoP* (Figure 5). Canker symptoms similar to those induced by the wild-type strain were observed on leaves inoculated with the complemented mutant strain, suggesting that PhoP is required for the virulence of *X. citri* subsp. *citri*.

**Analysis of RNA-seq data.** To clarify differences in gene expression between XHG3 and Δ*phoP*, transcriptome sequencing was performed. Total RNA integrity was detected by using an Agilent Technologies 2100 Bioanalyzer, and the degradation and contamination of RNA were analyzed by agarose gel electrophoresis (Appendix A). RNA purity was determined by Nanodrop spectrophotomete (OD_260/280_ ratio). All RNA evaluations meet the quality requirements for sequencing (Appendix A).

Sequencing reads were mapped against the reference genome. The XHG3 sample resulted in a total of 19,945,236 reads, with a total of 24,370,669 reads obtained for Δ*phoP* sample, indicating good data output quality (Appendix A).

In this study, a false discovery rate of 0.05, a (|log_2_(FoldChange)| > 1, and a q value < 0.005 were used as the cut-off for differential gene expression. There was a total of 1017 DGEs between Δ*phoP* and XHG3, among which 614 were up-regulated and 403 were down-regulated (Appendix A). Using a stringent *P*-value of > 0.005, 287 genes showing differential expression between XHG3 and Δ*phoP* were identified and then classified and enriched based on the GO database (http://www.geneontology.org/). The DEGs were significantly enriched in 20 GO terms (corrected_P-value < 0.1) involved in cell motility, localization of cell, flagellar motility, cellular components, bacterial-type flagellum and motor activity (Figure 6).

KEGG database (http://www.genome.jp/kegg/) analysis assigned pathways for 2733 genes, of which 481 showed significant differences in expression. These genes were significantly enriched (*P* < 0.05) in six signaling pathways, including flagellar assembly, two-component system, histidine metabolism, bacterial chemotaxis, ABC transporters, and bacterial secretion system (Appendix A).

**PhoP regulates the expression of T3SS-associated genes in *X. citri* subsp. *citri*.** Transcriptome analysis revealed that the complete T3SS-encoding *hrp* gene cluster, which contains 24 genes (*Xac0393* to *Xac0417*), including *hrpF*, *hpaB*, *hrpE, hrpD6, hrpD5, hpaA, hrcS, hrcR, hrcQ, hrcV, hrcU, hrpB1, hrpB2, hrcJ, hrpB4, hrpB5, hrcN, hrpB7, hrcT, hrcC* and *hpa1*, was down-regulated (expect for *Xac 0393* and *Xac 0417*), in Δ*phoP* compared with the wild-type strain (Table 2). Among 24 putative and known T3SS effectors in the *X. citri* subsp. *citri* genome [15], 15 effector genes, including *AvrBs2, HrpW (PopW), XopAD, XopAI, XopAK, XopE1, XopE3, XopI, XopK, XopN, XopQ, XopR, XopX, XopZ*, were down-regulated in Δ*phoP* (Table 3). These results suggested that PhoP positively regulates the expression of T3SS-associated genes.

**PhoP regulates the expression of most T4SS genes.** The genome of *X. citri* subsp. *citri* contains two T4SS clusters, one in the chromosome and the other on a plasmid [7]. We observed an increase in the expression of more than 10 genes (*virB4, virB1, virB11, virB10, virB8, virD4, XAC0096, XAC1918, XAC2609, XAC2610, phlA, XAC0323, HI* and *Cw1L*) coding for components of the T4SS or T4SS-interacting proteins [15] in the Δ*phoP* mutant (Table 4), indicating that PhoP negatively regulates theT4SS.

**PhoP positively regulates the expression of chemotaxis and flagella genes in *X. citri* subsp. *citri*.** A number of genes involved in motility and chemotaxis were regulated at a transcriptional level in the Δ*phoP* mutant. Twenty-five genes involved in flagellar assembly were distinctly down-regulated in the Δ*phoP* mutant compared with the wild-type, including *fliACEFGILMNPR, flhABF,* and *flgBCDEFGHIJKL*, all of which had log_2_.Fold_change values between −1.7 and −4.30. In addition, of the 17 bacterial chemotaxis genes, 16 of them showed decreased expression in the Δ*phoP*, including *cheW*, *cheA*, *cheY*, and *cheZ* (Table 5).

**PhoP regulates histidine metabolism.** Bacterial histidine metabolism involves 10 histidine biosynthesis genes coding for nine enzymes that catalyze 10 enzymatic reactions. Our result showed that eight of these histidine biosynthetic genes, *hisABCDFGHI* (*XAC1828* to *XAC1835*), were dramatically down-regulated in the Δ*phoP* mutant, The proteins encoded by these genes include 1-(5-phosphoribosyl)-5-[(5-phosphoribosylamino) methylideneamino] imidazole-4-carboxamide isomerase, histidine biosynthesis bifunctional protein, histidinol-phosphate aminotransferase, histidinol dehydrogenase, imidazole glycerol phosphate synthase subunit, imidazole glycerol phosphate synthase subunit and ATP phosphoribosyl transferase. In addition, the expression of both *hutH* and *hutU,* which are involved in the histidine utilization pathway, was also significantly decreased (Table 6).

**PhoP is involved in general metabolism and transport.** Many genes involved in general metabolism were regulated by PhoP, including *PstS, PstC, PstA* and *PstB*, all of which are involved in phosphate and amino acid transport. In addition, the expression of *XacPNP* (*Xac 2654*), encoding a plant natriuretic peptide (PNP)-like protein that is related to virulence in *X. citri* subsp. *citri* [35], was reduced in Δ*phoP*. In higher plants, PNPs elicit a number of responses that contribute to the regulation of homeostasis and growth [36], and help to induce the opening of stomata [37]. XacPNP shares significant sequence similarity and identical domain structure with other PNPs, and may also cause plant physiological responses such as stomatal opening [35].

**PhoP regulates the expression of some virulence-related genes.** To verify the RNA-Seq results and confirm the downstream genes regulated by PhoP, the expression of some target genes was examined using qRT-PCR -based analysis. The target genes included *hrpG, hrpX*, *hrcN* and *hrcQ* (T3SS regulators), *avrBs2* and *avrXacE1* (avirulence protein), *egl0028* (Cellulase), *cheA* and *cheY* (chemotaxis protein), *fliC* (flagellin), *flhF*(flagellar biosynthesis regulator), *pqqG* (pyrroloquinoline quinone biosynthesis protein), *rpoN* (RNA polymerase factor sigma-54), *XacPNP* (plant natriuretic peptide-like protein), *virB1* (T4SS protein) and *pthA* (T3SS effector). Results showed that the expression of *hrpG, hrpX*, *hrcN*, *hrcQ*, *avrBs2*, *avrXacE1*, *egl0028* (Cellulase), *cheA*, *cheY*, *fliC*, *flhF*, *pqqG*, *rpoN*, *XacPNP* and *pthA* was significantly lower in the Δ*phoP* mutant compared with XHG3, while the expression of *virB1* was increased (Figure 7), which was consistent with the RNA-sequencing results (Appendix A).

## 4. Discussion

PhoQ/PhoP is one of the best characterized TCS, a family of bacterial systems that sense environmental cues and effectors and trigger gene expression in response to these cues to enhance bacterial survival under stressful conditions or within host cells. The PhoQ/PhoP system has so far been characterized in *S. Typhimurium*, *Shigella* sp., *D. dadantii*, *X. oryzae* pv*. oryzae*, and *X. campestris* pv. *campestris* [18,20,22,23,38,39]. However, little is known about the PhoQ/PhoP proteins in the important plant pathogen *X. citri* subsp. *citri*. In this study, we investigated the potential contribution of PhoP to the virulence and fitness of *X. citri* subsp. *citri*, identified PhoP-regulated genes by RNA-seq analysis and demonstrated that PhoP is an important regulator involved in motility, biofilm formation, exoenzymes production and virulence of *X. citri* subsp. *citri* in citrus plants. In addition, we found that PhoP positively regulates the expression of genes involved in the T3SS, chemotaxis, flagella biosynthesis, and histidine metabolism in *X. citri* subsp. *citri.*

Chemotaxis is essential for the initial stages of bacterial infection, with several genes involved in chemotaxis signaling shown to be critical for entry of a pathogen into host cells [40]. Our study indicated that of the 17 bacterial chemotaxis genes, 16 of them showed decreased expression in the Δ*phoP* mutant, including *cheW*, *cheA*, *cheY* and *cheZ*.

In animal-pathogenic species *Yersinia enterocolitica and Shewanella oneidensis*, bacterial motility and biofilm formation are closely related to pathogenicity [41,42]. Flagella play a critical role in biofilm formation in *Y. enterocolitica*. Likewise, in plant-pathogenic bacterium, *X. campestris* pv. *campestris*, biofilm formation is controlled by cell-cell signaling and is required for full virulence in plant hosts [43], in *X. citri* subsp. *citri*, swimming motility, biofilm formation is required for canker development [28,44]. Our RNA-seq analysis showed that 25 genes involved in flagellar assembly, including *fliACEFGILMNPR*, *flhABF*, and *flgBCDEFGHIJKL*, had distinctly lower levels of expression in Δ*phoP* compared with the wild-type strain. Although primarily associated with cell movement, flagella also act as virulence determinants [44]. The expression of chemotaxis and motility genes by *X. oryzae* pv. *oryzae* is required for entry, colonization, and virulence in the host [45]. In the current study, the Δ*phoP* mutant showed a significant reduction in motility and biofilm formation compared with wide-type strain XHG3, further qRT-PCR analysis further confirmed the decreased expression of flagella- and chemotaxis-associated genes *fliC*, *flhF*, *cheA*, *cheY*, and *rpoN* in Δ*phoP*. Therefore, we concluded that the observed repression of motility and biofilm formation in the mutant strain results from the altered expression of genes involved in chemotaxis and flagellar biosynthesis.

*X. citri* subsp. *citri* uses several mechanisms to colonize host plants. These include the T3SS, which delivers virulence effector proteins [9,10,46] and the type II secretion system (T2SS), which is thought to be involved in canker development [11,47,48]. Mutation of *xpsD*, coding for a component of the T2SS, hinders the secretions of cellulase, protease, and amylase, and affects the ability of *X. citri* subsp. *citri* to colonize tissues and hydrolyze cellulose [47]. BglC3, an extracellular endoglucanase, is necessary for the full virulence of *X.citri* subsp. *citri* [12]. Extracellular enzymes have been extensively characterized in *Pectobacterium* and *Dickeya* species because of their essential functions in the development of soft rot symptoms [49]. Our results showed that the activities of extracellular enzymes such as polygalacturonase (PG), cellulase and amylase were significantly decreased in Δ*phoP*. In addition, the Δ*phoP* mutant demonstrated significantly reduced virulence compared with the wild-type strain when inoculated onto citrus leaves. Using qRT-PCR analysis, we also showed that the expression of *egl0028*, encoding cellulose, was reduced in Δ*phoP*.

The T3SS plays an important role in bacterial pathogenicity. It is encoded by a cluster of hypersensitive response and pathogenicity (*hrp*) genes that are critical for bacterial virulence. The T3SS translocates effector proteins into plant cells, where they either suppress the host defense system or interfere with host cellular processes [12]. The expression of *hrp* genes is controlled by two regulators HrpG and HrpX, which are important pathogenicity regulators in *X. citri* subsp. *citri* [8,50]. The transcriptome analysis carried out in the current study showed that PhoP positively regulates many genes involved in the T3SS, including 22 *hrp* genes and 15 putative and known T3SS effectors, while further qRT-PCR analysis confirmed that PhoP positively regulates the expression of *hrpG*, *hrpX*, *hrcN*, *hrcQ*, and *pthA* (T3SS effector). PhoQ/PhoP is also required for *hrpG* expression and virulence in *X. oryzae* pv. *oryzae* [23]. Accordingly, we believe that PhoP can influence the pathogenicity of *X. citri* subsp. *citri* through the expression of *hrp* genes and T3SS effectors.

RNA-Seq and qRT-PCR analyses also confirmed that *XacPNP* (*Xac2654*), which encodes a PNP-like protein, was positively regulated by PhoP. PNPs contributed to the regulation of homeostasis reponses and growth. XacPNP plays an important role in the infection process by modifying host responses to create favorable conditions for *X. citri* subsp. *citri* growth [35]. Therefore, we can infer that XacPNP is involved in PhoP-mediated regulation of pathogenicity. In addition to altering plant defenses, *X. citri* subsp. *citri* must adapt its metabolism to the nutrient-poor and toxin-laden (either preformed or induced) intercellular spaces of host cells [51], which form part of the host defense responses [52,53]. Our transcriptomic analysis indicated that PhoP alters the expression of many genes involved in metabolic processes. For example, in the histidine metabolism, our results showed that eight histidine biosynthetic genes (*hisABCDFGHI*) were dramatically down-regulated in Δ*phoP*, while the expressions of both *hutH* and *hutU*, which are involved in the histidine utilization pathway, was also significantly decreased. Histidine biosynthesis requires 10 enzymatic reactions involving proteins encoded by seven bacterial genes, Imidazoleglycerol-phosphate dehydratase (IGPD) catalyzes the sixth step in the histidine biosynthesis pathway, and is the first identified enzyme exclusively dedicated to histidine biosynthesis in bacteria. Mutations in *hisB* and *IGPD* inhibit biofilm formation in *Staphylococcus xylosus* [54], while mutations within histidine metabolism genes *hisD*, *FhisF, hisG* and *hut*G lead to decreased virulence of *Pseudomonas savastanoi* pv. *savastanoi* in olive [55]. Further, a recent study showed that the histidine utilization (Hut) pathway is involved in quorum sensing and contributes to virulence in *X. oryzae* pv. *oryzae*, as *X. oryzae* pv. *oryzae* Δ*hutG* and Δ*hutU* deletion mutants showed reduced virulence [56]. Therefore, we deduce that histidine metabolism may also be necessary for the virulence of *X. citri* subsp. *citri*.

T4SS play a fundamental role in disease progression in several important animal and human pathogens [57] as well as in plant pathogens *Agrobacterium tumefaciens* [58] and *Erwinia* sp. [59]. However, our findings showed that 11 T4SS genes and 13 genes coding for T4SS-interacting proteins were up-regulated in Δ*phoP*. Thus, exactly how PhoP affects the expression of T4SS genes is not yet clear, and requires further investigation. 

In plant-pathogenic bacteria, PhoP controls a key aspect of *X*. *oryzae* pv. oryzae AvrXA21 virulence through regulation of *hrpG* [23]. Among all PhoP-regulated genes, 137 genes were down-regulated while 77 genes were upregulated in the *phoP* knockout mutant of *X*. *oryzae* pv. oryzae PXO99A, which encode primarily hypothetical proteins or proteins associated with the cell envelope, protein fate, regulatory functions, or transport and binding [24]. In *Dickeya dadantii* 3937, a mutation in *phoQ* affects transcription of at least 40 genes. However, there was no investigation involving PhoP [22]. Our results in *X.citri* subsp. *citri* show that PhoP regulates more genes and has more extensive functions. Therefore, we may conclude that PhoP is a global regulatory factor in *X. citri* subsp. *citri*, affecting the pathogenicity and virulence of this important citrus pathogen by regulating chemotaxis and motility, biofilm formation, T3SS proteins, histidine biosynthesis, and the production of extracellular enzymes. However, further investigation of these regulatory mechanisms is needed.

## Figures and Tables

**Figure 1 genes-10-00340-f001:**
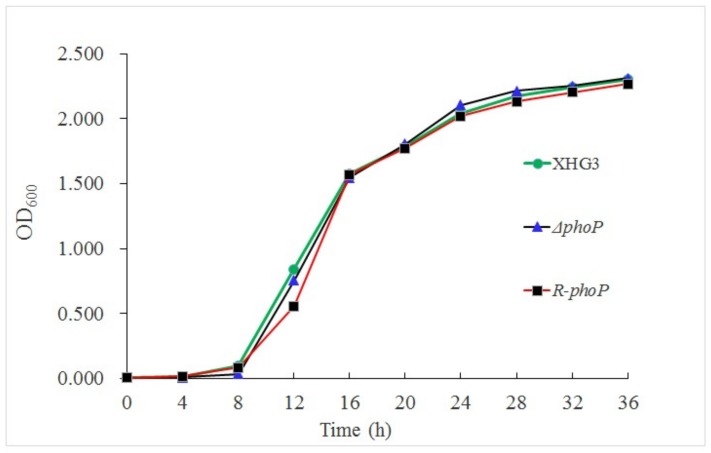
Growth curves of wild-type strain XHG3, mutant strain Δ*phoP* and complementation strain R-*phoP*. The growth (OD_600_) of each strain in YEB broth at 30 °C was measured at 2-h intervals. Data points represent the means of three biological replicates.

**Figure 2 genes-10-00340-f002:**
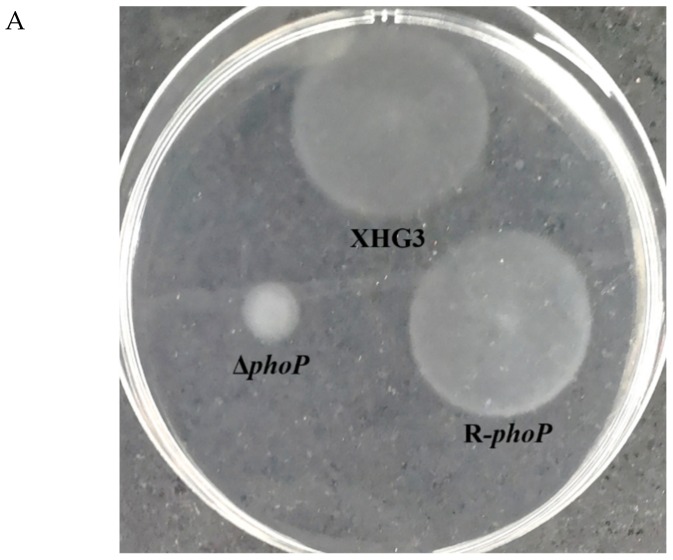
Motility assay for wild-type strain XHG3, mutant strain Δ*phoP*, and complementation strain R-*phoP*. Aliquots (2 μL) of each of the bacterial suspensions were inoculated onto the surface of semi-solid agar medium plate and incubated for 72 h at 28 °C to assess the bacterial motility. (**A**) the Δ*phoP* mutant showed considerably reduced motility compared with wild-type strain XHG3; (**B**) with a 70.6% reduction in colony diameter compared with that of strain XHG3.

**Figure 3 genes-10-00340-f003:**
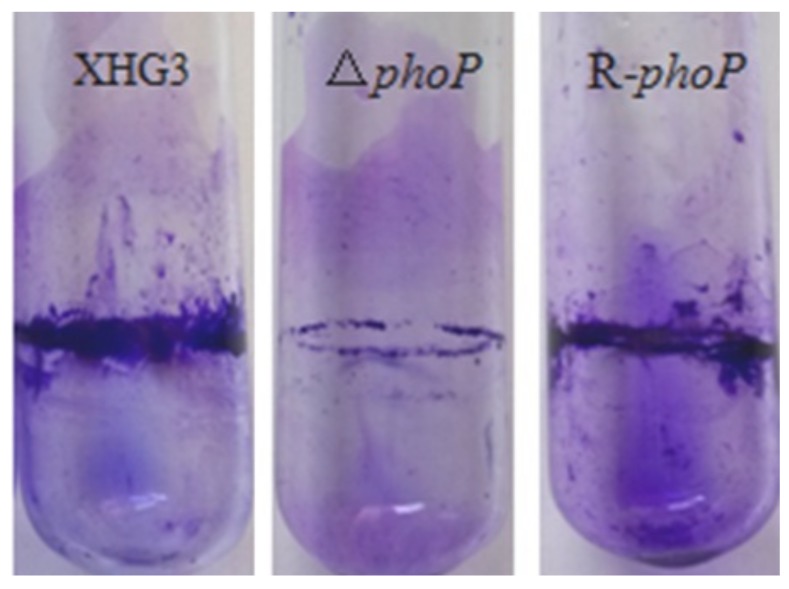
Biofilm formation by wild-type strain XHG3, the Δ*phoP* mutant and complementation strain R-*phoP*. A 10-μL aliquot of bacterial cell culture (OD_600_ = 0.5) was inoculated into tubes containing SONG medium and incubated for 72 h at 28 °C.

**Figure 4 genes-10-00340-f004:**
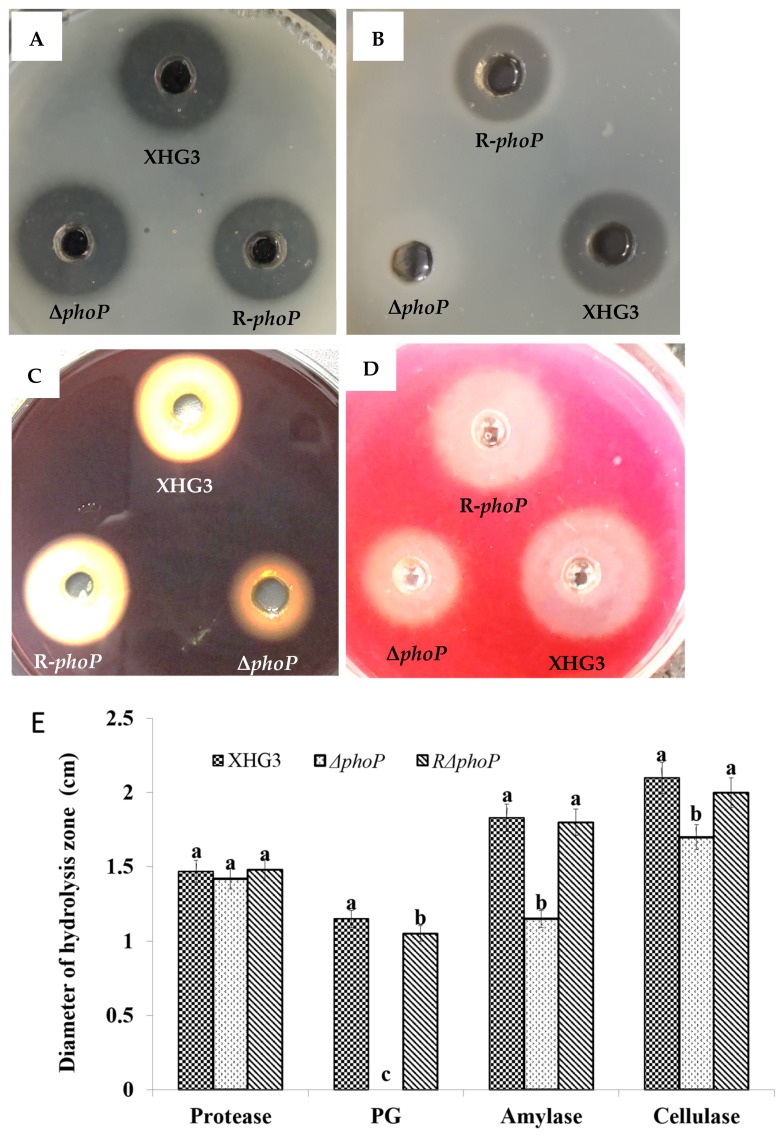
Determination of extracellular enzymes produced by wild-type strain XHG3, the Δ*phoP* mutant and complementation strain R-*phoP*. Thirty-microliter aliquots of bacterial supernatants were added to the wells of the exoenzyme test plates for further cultivation at 28 °C for 24 h. Production of hydrolysis zones by protease (**A**), polygalacturonase (PG) (**B**), amylase (**C**) and cellulase (**D**) was assessed on plates containing skimmed milk (**A**), polygalacturonate (**B**), starch (**C**) and sodium carboxymethyl cellulose (**D**), respectively. Bar graph (**E**) showing the diameters of hydrolysis zones from Figure 4A–D. Values are the means of three replicates ± standard deviation.

**Figure 5 genes-10-00340-f005:**
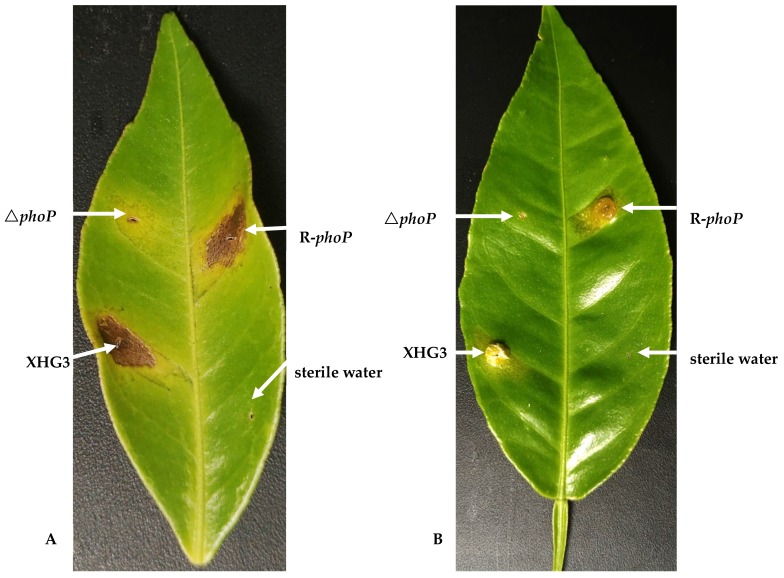
Pathogenicity assay for wild-type strain XHG3, the Δ*phoP* mutant, and complementation strain R-*phoP* on citrus leaves. The leaves of two citrus cultivars, Xinhuigan (**A**) and Orah (**B**), were inoculated with bacterial suspension (10^7^ CFU/mL) over 3 days using needleless syringes. Leaves inoculated with Δ*phoP* showed slight canker symptoms, while obvious canker symptoms were observed on leaves inoculated with the wild-type or complementation strains.

**Figure 6 genes-10-00340-f006:**
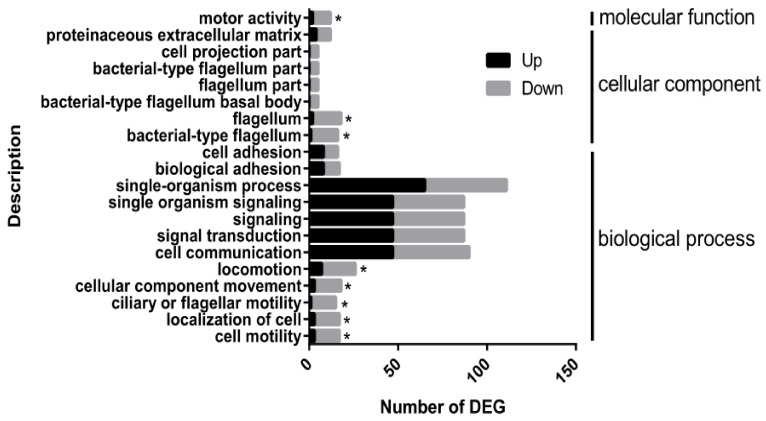
Distribution of differentially-expressed genes (DEGs) between XHG3 and Δ*phoP* by enriched gene ontology (GO) terms. DEGs in Δ*phoP* are annotated by GO. The 20 most enriched GO terms are shown. Significantly over-represented GO terms with a corrected P-value of <0.05 are marked with an asterisk.

**Figure 7 genes-10-00340-f007:**
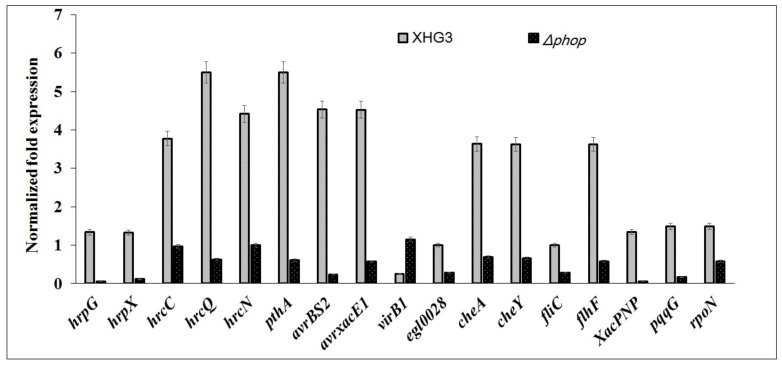
Relative expression of genes associated with the virulence of wild-type strain XHG3 and mutant strain Δ*phoP* as determined by qRT-PCR analysis. The 16S rRNA gene was used as an endogenous control to normalize gene expression data. Target genes were: *hrpG*, *hrpX*, *hrcN*, and *hrcQ* (T3SS regulators), *avrBs2* and *avrXacE1* (avirulence proteins), *egl0028* (cellulase), *cheA* and *cheY* (chemotaxis proteins), *fliC* (flagellin), *flhF* (flagellar biosynthesis regulator), *pqqG* (pyrroloquinoline quinone biosynthesis protein), *rpoN* (RNA polymerase sigma factor 54), *XacPNP* (plant natriuretic peptide-like protein), *virB1* (T4SS protein), *pthA* (T3SS effector).

**Table 1 genes-10-00340-t001:** Bacterial strains and plasmids.

Strain and Plasmid	Relevant Characteristics	Origin
*X. citri* subsp. *citri*		
XHG3	Wild-type strain	This study
Δ*phoP*	*phoP* knock-out mutant of XHG3	This study
R-*phoP*	complementation strain of Δ*phoP*	This study
*E. coli*		
K12 CC118	*gyrA*,*recA*,*λ pir*	Lab collection
DH5α	*deoR*,*recA*,*endA*,*hsdR*,*supE*,*thi*,*gyrA*,*relA*	Lab collection
Plasmids		
pKNG101	Sm^R^,*SacB*,*mobRK2*,*oriR6K* (pir-minus)	Lab collection
pKNG-Δ*phoP*	*PhoP* knock-out fragment ligated on pKNG101	This study
pBBR1*- phoP*	*PhoP* complementation fragment ligated on pBBR1MCS4	This study

**Table 2 genes-10-00340-t002:** Comparison of T3SS genes cluster expression in the XHG3 and Δ*phoP.*

Gene	Locus tag	log_2_.Fold_change.	q-value
*hrpF*	XAC0394	−1.77	6.33 × 10^−14^
*hpaB*	XAC0396	−3.20	0
*hrpE*	XAC0397	−5.16	7.95 × 10^−4^
*hrpD6*	XAC0398	−1.95	8.94 × 10^−7^
*hrpD5*	XAC0399	−2.82	1.82 × 10^−5^
*hpaA*	XAC0400	−2.46	2.68 × 10^−3^
*hrcS*	XAC0401	−3.86	1.11 × 10^−17^
*hrcR*	XAC0403	−4.39	2.06 × 10^−16^
*hrcQ*	XAC0404	−4.26	1.14 × 10^−12^
*hrcV*	XAC0405	−2.03	7.26 × 10^−12^
*hrcU*	XAC0406	−2.99	8.22 × 10^−28^
*hrpB1*	XAC0407	−4.21	3.45 × 10^−12^
*hrpB2*	XAC0408	−4.02	6.59 × 10^−20^
*hrcJ*	XAC0409	−4.48	1.95 × 10^−12^
*hrpB4*	XAC0410	−3.03	6.46 × 10^−7^
*hrpB5*	XAC0411	−4.05	6.51 × 10^−9^
*hrcN*	XAC0412	−3.08	2.02 × 10^−4^
*hrpB7*	XAC0413	−3.76	2.94 × 10^−6^
*hrcT*	XAC0414	−2.85	5.01 × 10^−7^
*hrcC*	XAC0415	−1.69	3.18 × 10^−183^
*hpa1*	XAC0416	−-6.06	6.33 × 10^−14^

log_2_.Fold_change: log_2_ (readcount_**Δ***phoP* / readcount_XHG3).

**Table 3 genes-10-00340-t003:** Gene expression of T3SS effectors between Δ*phoP* and XHG3.

Effector family	Locus tag	log_2_.Fold_change.	q-value
AvrBs2	XAC0076	−1.82	5.36 × 10^−8^
AvrBs3(PthA1)	XACa002	-	-
AvrBs3(PthA2)	XACa0039	-	-
AvrBs3(PthA3)	XACb0015	-	-
AvrBs3(PthA4)	XACb0065	-	-
HrpW (PopW)	XAC2922	−2.52	2.41 × 10^−55^
XopAD (Skwp, RSc3401)	XAC4213	−2.13	2.92 × 10^−44^
XopAE (HpaF/G/PopC)	XAC0393	N	N
XopAI (HopO1 (HopPto, HopPtoS), HopAI1 (HolPtoAI))	XAC3230	−1.99	2.05 × 10^−26^
XopAK (HopAK1 (HopPtoK, HolPtoAB) C terminal domain	XAC3666	−2.05	2.38 × 10^−7^
XopE1 (AvrXacE1, HopX, AvrPPhE)	XAC0286	−1.79	1.61 × 10^−17^
XopE2 (AvrXacE3, AvrXccE1)	XACb0011	N	N
XopE3 (AvrXacE2, HopX, AvrPPhE)	XAC3224	−1.11	2.24 × 10^−3^
XopF2	XAC2785	N	N
XopI	XAC0754	−2.35	1.60 × 10^−7^
XopK	XAC3085	−3.20	1.40 × 10^−32^
XopL	XAC3090	N	N
XopN (HopAU1)	XAC2786	−2.28	6.35 × 10^−23^
XopP	XAC1208	−1.24	8.87 × 10^−4^
XopQ (HopQ1)	XAC4333	−3.84	5.58 × 10^−19^
XopR	XAC0277	−2.61	4.79 × 10^−25^
XopV	XAC0601	N	N
XopX (HolPsyAE)	XAC0543	−2.79	7.16 × 10^−51^
XopZ (HopAS, AWR)	XAC2009XAC2990	−2.35N	6.46 × 10^−19^N

Note: -, do not show in the RNA-Seq; N, not considered expression difference, log_2_. Fold_change: log_2_ (read count_Δ*phoP* / read count_XHG3).

**Table 4 genes-10-00340-t004:** Gene expression of T4SS between Δ*phoP* and XHG3.

Gene	Locus tag	log_2_.Fold_change.	q-value
*virB4*	XAC2614	2.34	1.57 × 10^−258^
*virB1*	XAC2617	2.90	8.15 × 10^−188^
*virB11*	XAC2618	2.18	5.55 × 10^−99^
*virB10*	XAC2619	2.19	6.58 × 10^−139^
*virB8*	XAC2621	1.86	2.10 × 10^−93^
virD4	XAC2623	1.94	1.22 × 10^−115^
*XAC0096*	XAC0096	2.99	6.06 × 10^−134^
*XAC1918*	XAC1918	1.62	1.14 × 10^−12^
*XAC2609*	XAC2609	1.77	2.55 × 10^−135^
*XAC2610*	XAC2610	1.69	3.98 × 10^−128^
*phlA*	XAC2885	2.04	1.60 × 10^−42^
*XAC0323*	XAC0323	2.12	2.22 × 10^−26^
*HI*	XAC0466	2.22	8.25 × 10^−106^
*Cw1L*	XAC3634	2.1	9.38 × 10^−43^

log_2_.Fold_change: log_2_ (read count_Δ*phoP*/read count_XHG3).

**Table 5 genes-10-00340-t005:** Expression of flagellar assembly and bacterial chemotaxis genes in Δ*phoP*.

Gene_id	log_2_.Fold_change	*P*-value	Gene Description
XAC2447	−2.7359	1.47 × 10^−4^	Chemotaxis protein CheW/*cheW*
XAC2448	−2.908	2.51 × 10^−16^	Methyl-accepting chemotaxis protein I/*tsr*
XAC3213	−1.4002	4.098 × 10^−3^	Methyl-accepting chemotaxis protein I/GN=*tsr*
XAC2866	−1.6455	0	Methyl-accepting chemotaxis citrate transducer/*tcp*
XAC1894	−1.6016	9.78 × 10^−5^	Methyl-accepting chemotaxis protein I/*tsr*
XAC1899	−1.257	2.948 × 10^−3^	Methyl-accepting chemotaxis protein I/*tsr*
XAC1892	−1.9605	2.22 × 10^−5^	Methyl-accepting chemotaxis protein I/*tsr*
XAC1895	1.37	2.06 × 10^−8^	Methyl-accepting chemotaxis serine transducer/
XAC1930	−2.3919	9.80 × 10^−19^	Chemotaxis protein CheA/*cheA*
XAC1931	−2.147	6.31 × 10^−9^	Protein phosphatase CheZ/*cheZ*
XAC1932	−2.0968	3.54 × 10^−8^	Chemotaxis protein CheY/*cheY*
XAC1933	−2.6854	7.24 × 10^−12^	RNA polymerase sigma factor for flagellar operon/*fliA*
XAC1934	−2.3675	3.02 × 10^−10^	Flagellum site-determining protein YlxH/*ylxH*
XAC1935	−4.4217	6.82 × 10^−24^	Flagellar biosynthesis protein FlhF/*flhF*
XAC1936	−2.9695	6.93 × 10^−18^	Flagellar biosynthesis protein FlhA/*flhA*
XAC1937	−3.0715	2.47 × 10^−5^	Flagellar biosynthetic protein FlhB/*flhB*
XAC1938	−1.8277	1.47 × 10^−5^	Uncharacterized signaling protein PA1727/PA1727//0
XAC1940	−2.2302	9.73 × 10^−10^	Uncharacterized signaling protein CC_0091/GN=CC_0091
XAC1941	−2.2199	1.924 × 10^−3^	Flagellar biosynthetic protein FliR/*fliR*
XAC1942	−2.7865	3.83 × 10^−4^	-//-
XAC1944	−1.7605	3.91 × 10^−9^	Flagellar biosynthetic protein FliP/*fliP*
XAC1945	−3.5139	1.58 × 10^−6^	-//-
XAC1946	−3.9225	0.853 × 10^−3^	Flagellar motor switch protein FliN/*fliN*
XAC1947	−3.492	1.35 × 10^−8^	Flagellar motor switch protein FliM/*fliM*
XAC1948	−2.9248	1.76 × 10^−8^	Flagellar protein FliL/GN=*fliL*
XAC1949	−2.264	1.97 × 10^−26^	-//-
XAC1950	−1.583	8.74 × 10^−4^	-//-
XAC1951	−2.256	1.43 × 10^−17^	Flagellum-specific ATP synthase/*fliI*
XAC1952	−2.2921	7.37 × 10^−9^	-//-
XAC1953	−1.5637	1.43 × 10^−8^	Flagellar motor switch protein FliG/*fliG*
XAC1954	−2.7658	7.29 × 10^−20^	Flagellar M-ring protein/*fliF*
XAC1955	−4.3033	2.79 × 10^−6^	Flagellar hook-basal body complex protein FliE/*fliE*
XAC1969	−1.3085	4.57 × 10^−35^	RNA polymerase sigma-54 factor/rpoN
XAC1975	−1.699	1.03 × 10^−45^	A-type flagellin/*fliC*
XAC1976	−2.816	7.13 × 10^−19^	Flagellar hook-associated protein/*flgL*
XAC1977	−3.0454	1.74 × 10^−23^	Flagellar hook-associated protein/*flgK*
XAC1978	−3.1714	9.31 × 10^−16^	Peptidoglycan hydrolase FlgJ/*flgJ*
XAC1979	−3.7131	1.32 × 10^−14^	Flagellar P-ring protein/*flgI*
XAC1980	−2.9579	1.89 × 10^−8^	Flagellar L-ring protein/*flgH*
XAC1981	−3.7208	9.54 × 10^−24^	Flagellar basal-body rod protein FlgG/*flgG*
XAC1982	−2.1116	1.26 × 10^−8^	Flagellar basal-body rod protein FlgF/*flgF*
XAC1983	−2.751	2.42 × 10^−22^	Flagellar hook protein FlgE/*flgE*
XAC1984	−3.135	3.25 × 10^−11^	Basal-body rod modification protein FlgD/*flgD*
XAC1985	−3.1884	2.24 × 10^−4^	Flagellar basal-body rod protein FlgC/*flgC*
XAC1986	−3.7208	1.21 × 10^−12^	Flagellar basal body rod protein FlgB/*flgB*
XAC1988	−2.8726	1.54 × 10^−6^	-//-

log_2_.Fold_change: log_2_ (read count_Δ*phoP*/read count_XHG3).

**Table 6 genes-10-00340-t006:** Gene expression in histidine metabolism in Δ*phoP*.

Gene_id	log_2_.Fold_change	*P*-value	Gene Description
XAC1637	−1.4138	4.91 × 10^−7^	*HutH*/Histidine ammonia-lyase
XAC1635	−1.8604	9.51 × 10^−8^	HutU/Urocanate hydratase
XAC1834	−2.8393	3.05 × 10^−17^	*HisF*/Imidazole glycerol phosphate synthase subunit
XAC1833	−2.906	2.31 × 10^−20^	*HisA*/1-(5-phosphoribosyl)-5-[(5-phosphoribosylamino)methylideneamino] imidazole-4-carboxamide isomerase
XAC1835	−2.9827	3.98 × 10^−21^	*HisIE*/Histidine biosynthesis bifunctional protein
XAC1832	−3.343	3.05 × 10^−13^	*HisH*/Imidazole glycerol phosphate synthase subunit
XAC1831	−3.3447	5.44 × 10^−70^	*HisB*/Histidine biosynthesis bifunctional protein
XAC1829	−3.4986	8.67 × 10^−60^	*HisD*/Histidinol dehydrogenase
XAC1830	−3.506	3.44 × 10^−40^	*HisC*/Histidinol-phosphate aminotransferase
XAC1828	−3.7577	7.18 × 10^−43^	*HisG*/ATP phosphoribosyltransferase

log_2_.Fold_change: log_2_ (read count_Δ*phoP*/read count_XHG3).

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
