# Peer review of "Global Regulator PhoP is Necessary for Motility, Biofilm Formation, Exoenzyme Production, and Virulence of *Xanthomonas citri* Subsp. *citri* on Citrus Plants"

_genes, 2019, doi:10.3390/genes10050340_

Round 1
Reviewer 1 Report
Authors have investigated the role of PhoP in the citrus pathogen X. citri subsp. citri (Xcc)using a phoP deletion mutant. Their results reveal that the phoP mutant showed significantly decreased virulence on citrus leaves, reduction in cell motility and biofilm formation as well as decreased levels of cellulase, amylase, and polygalacturonase enzymes in Xcc. In order to better understand the Xccvirulence mechanisms, they have used high-throughput RNA sequencing technology to compare the transcriptomes between the wild-type and phoPmutant strains. Their analysis revealed 1017 differentially-expressed genes, of which 614 genes were up-regulated and 403 genes were down-regulated in the mutant strain, respectively. Bioinformatic analysis allowed them to conclude that PhoP regulated theexpression of a large set of virulence genes, including type III and IV secretion system as well as l genes involved in chemotaxis, flagellar,and histidine biosynthesis.
Major concerns:
As described by the authors, the PhoQ/PhoP system has been characterized in many pathogenic bacteria including Salmonella typhimurium, Shigella sp.,Dickeya dadantii, Xanthomonas oryzae pv. oryzae, and Xanthomonas campestris pv. campestris. However, the effects and regulatory mechanisms of PhoQ/PhoP may differamongst pathogenic bacteria, even those belonging to the same genus. Therefore, it is worth with the characterization of PhoQ/PhoP system in Xcc, a serious pathogen of citrus plants. Unfortunately, the authors did not discuss much about the common and different regulations of PhoP among Xcc and other pathogenic bacteria.
The preparation of complementation strain R-phoP was not clearly or completely described. It is also not clear how and when the complement PhoP was expressed in all the experiments.
Minor concerns:
The manuscript has too many careless typo (specially with spaces between sentences or words). If the authors performed the experiments and data analysis with the same careless attitude, the integrity and quality of their results are in serious trouble.
Fig. 2A is missing the labels for the wild-type strain XHG3, mutant strain ΔphoP, and complementation strain R-phoP. Labels should be placed side-by-side with the correspondent colony.
What is CK in Fig. 5?
Fig. 7. This graph has a grid and a grid was not shown in graph 4E. In the Y axis should be “normalized fold expression” instead “normalized ford expression”
Table 1. In the “strain and plasmid” column; E. coli K12 CC118 and Plasmids pKNG101 should be in a single line for better visualization. Apparently, an autofit will be working nicely. In column “origin”, the last “This study” is not at the center as the others are.
Table 4. The font for XAC0323 and Cw1L in the last two columns is different.
Table 5 is missing a p-value for XAC2866.
Table 6: Table title has different font sizes. Column width should be optimized.
Fig. S1: A diagram of the cloning mechanism will help to explain the band patterns.
What are the sizes of all bands in Fig. S2?
Fig. S3: The gene labels should be above the X-axis instead of overlapping with the data bars.
Table S1: Table spacing is different from others. This table is long and should remove large blank spaces between lines.
Table S2: Table spacing is different from others. Different fonts were detected in the table footnote.
Table S3: Table has a different format from others, such fonts and table grid.
Table S4: Table has a different format from others, such fonts and table grid. Column width must be adjusted for a better content fitting. In gene ID column, “xac” is showed in lowercase but is showed uppercase “XAC” in Table S3.
Author Response
Responses to the reviewer’s comments
Comments from Reviewer: 1:
Authors have investigated the role of PhoP in the citrus pathogen X. citri subsp. citri (Xcc) using a phoP deletion mutant. Their results reveal that the phoP mutant showed significantly decreased virulence on citrus leaves, reduction in cell motility and biofilm formation as well as decreased levels of cellulase, amylase, and polygalacturonase enzymes in Xcc. In order to better understand the Xcc virulence mechanisms, they have used high-throughput RNA sequencing technology to compare the transcriptomes between the wild-type and phoP mutant strains. Their analysis revealed 1017 differentially-expressed genes, of which 614 genes were up-regulated and 403 genes were down-regulated in the mutant strain, respectively. Bioinformatic analysis allowed them to conclude that PhoP regulated the expression of a large set of virulence genes, including type III and IV secretion system as well as l genes involved in chemotaxis, flagellar, and histidine biosynthesis.
Major concerns:
As described by the authors, the PhoQ/PhoP system has been characterized in many pathogenic bacteria including Salmonella typhimurium, Shigella sp., Dickeya dadantii,Xanthomonas oryzae pv. oryzae, and Xanthomonas campestris pv. campestris.However, the effects and regulatory mechanisms of PhoQ/PhoP may differ amongst pathogenic bacteria, even those belonging to the same genus.Therefore, it is worth with the characterization of PhoQ/PhoP system inXcc, a serious pathogen of citrus plants.Unfortunately, the authors did not discuss much about the common and different regulations of PhoP among Xcc and other pathogenic bacteria.
Response: Changed as suggested
PhoQ/PhoP system has so far been best characterized in animal-pathogenic bacterium, but its research in plant-pathogenic bacterium is still not complete and thorough enough. At the end of the manuscript discussion, we added comparisons with other plant bacteria, and believe that PhoP regulates more genes and has more extensive functions in X.citri subsp. citris.
The preparation of complementation strain R-phoP was not clearly or completely described. It is also not clear how and when the complement PhoP was expressed in all the experiments.
Response: The complete complement construction method has been supplemented in the method. Because all the biological assays showed that the complementation strain R-phoP were similar to the wild type strain, transcriptome and qRT-PCT did not include the complementation strain. complementation strain was not the focus of this study, and the purpose of our study was to determine the function of PhoP mutation.
Minor concerns:
The manuscript has too many careless typo (specially with spaces between sentences or words). If the authors performed the experiments and data analysis with the same careless attitude, the integrity and quality of their results are in serious trouble.
Response: Thanks for the reviewer’s criticism. This is our carelessness in writing, but we are rigorous in doing experiments
Fig. 2A is missing the labels for the wild-type strain XHG3, mutant strain ΔphoP, and complementation strain R-phoP. Labels should be placed side-by-side with the correspondent colony.
Response: Changed as suggested
What is CK in Fig. 5?
Response: Changed as suggested (CK is sterile water)
Fig. 7. This graph has a grid and a grid was not shown in graph 4E.In the Y axis should be “normalized fold expression” instead “normalized ford expression”
Response: Changed as suggested
Table 1. In the “strain and plasmid” column; E. coli K12 CC118 and Plasmids pKNG101 should be in a single line for better visualization. Apparently, an autofit will be working nicely. In column “origin”, the last “This study” is not at the center as the others are.
Response: Changed as suggested
Table 4. The font for XAC0323 and Cw1L in the last two columns is different.
Response: Changed as suggested
Table 5 is missing a p-value for XAC2866.
Response: Changed as suggested
Table 6: Table title has different font sizes. Column width should be optimized.
Response: Changed as suggested
Fig. S1: A diagram of the cloning mechanism will help to explain the band patterns.
Response: Changed as suggested
What are the sizes of all bands in Fig. S2?
Response: Changed as suggested
Fig. S3: The gene labels should be above the X-axis instead of overlapping with the data bars.
Response: Changed as suggested
Table S1: Table spacing is different from others. This table is long and should remove large blank spaces between lines.
Response: Changed as suggested
Table S2: Table spacing is different from others. Different fonts were detected in the table footnote.
Response: Changed as suggested
Table S3: Table has a different format from others, such fonts and table grid.
Response: Changed as suggested
Table S4: Table has a different format from others, such fonts and table grid. Column width must be adjusted for a better content fitting. In gene ID column, “xac” is showed in lowercase but is showed uppercase “XAC” in Table S3.
Response: Changed as suggested

Reviewer 2 Report
The work by Wei and co-workers describes the phenotype of a Xanthomonas citri delta phoP mutant, and specially, the impact of this mutation in the pathogenicity of this important plant pathogen. It was a well conducted exploratory research, showing a nice picture of this mutant by RNAseq. However, authors have to be cautious to discuss their data, and to draw conclusions, since they examined the bacterial RNA content by growing cells in liquid media only.
PhoP is a master regulator that operates in the adaptation of the bacterium to a multitude of environments. Such a master regulator probably has a more global function, other than solely to control disease in X. citri. In this sense, the perception of the authors was fair and correct throughout the ms.
Discussion lines 404-418, authors forgot to mention the work of Rigano et al, and Malamud et al, who were the pioneers to say and show that reduced motility/biofilm formation impact on pathogenicity…it has to be considered.
Major points:
Line 251, in the description of Fig. 4, authors mixed up everything, which makes it difficult for the reader to understand them, and therefore to judge, what they are trying to say. Figure legend and text are different.
Lines 269, and fig. 5: first of all, if the delta-phoP mutant has a clear phenotype of decreased motility, and decreased biofilm formation, one would expect it to be impaired in plant colonization- so, the obvious test to be carried out here would be aspersion inoculation. Aspersion would show that mutant probably cannot infect the host, because it has reduced motility and biofilm. This idea needs to be addressed in the ms.
Results and discussion (line 400), amongst the genes down-regulated in the delta-phoP mutant are some involved with chemotaxis – this is another reason why infection by spray should have been conducted
Fig. 5- what does CK mean?
Minor points:
- Be consistent with the abbreviation of X. citri throughout the text
- Line 60, give the reference for article describing the genome of X. citri
- Fig. 2A, although we can identify the mutant, I suggest labeling the spots to undoubtedly demonstrate what is what; e.g. wild type, delta-phoP, and complemented mutant
- Line 370- “avirulence protein”
Author Response
Comments from Reviewer: 2:
Comments and Suggestions for Authors
The work by Wei and co-workers describes the phenotype of a Xanthomonas citri delta phoP mutant, and specially, the impact of this mutation in the pathogenicity of this important plant pathogen. It was a well conducted exploratory research, showing a nice picture of this mutant by RNAseq. However, authors have to be cautious to discuss their data, and to draw conclusions, since they examined the bacterial RNA content by growing cells in liquid media only.
Response: Thank the reviewer suggestions. We made individual modifications.
PhoP is a master regulator that operates in the adaptation of the bacterium to a multitude of environments. Such a master regulator probably has a more global function, other than solely to control disease in X. citri. In this sense, the perception of the authors was fair and correct throughout the ms.
Response: We agree with the reviewer’s comments
Discussion lines 404-418, authors forgot to mention the work of Rigano et al, and Malamud et al, who were the pioneers to say and show that reduced motility/biofilm formation impact on pathogenicity…it has to be considered.
Response: Changed as suggested
Major points:
Line 251, in the description of Fig. 4, authors mixed up everything, which makes it difficult for the reader to understand them, and therefore to judge, what they are trying to say. Figure legend and text are different.
Response: Changed as suggested
Lines 269, and fig. 5: first of all, if the delta-phoP mutant has a clear phenotype of decreased motility, and decreased biofilm formation, one would expect it to be impaired in plant colonization- so, the obvious test to be carried out here would be aspersion inoculation. Aspersion would show that mutant probably cannot infect the host, because it has reduced motility and biofilm. This idea needs to be addressed in the ms.
Response: PhoP is a globe regulatory factors, motility and biofilm regulation is a part of PhoP function. we inoculated Xcc suspension by a syringe without a needle to make extrusion injection into the citrus leaves (with wound), therefore, we think the reduced virulence of delta-phoP mutant is not entirely due to the motility and biofilm reduction, it may be involved in T3SS and effectors.
Results and discussion (line 400), amongst the genes down-regulated in the delta-phoP mutant are some involved with chemotaxis – this is another reason why infection by spray should have been conducted
Response: X. citri subsp. citri invades leaf mainly from the wound, and we have ever done by spray inoculation, which is difficult to appear the canker symptoms, even if the wild type strain.
Fig. 5- what does CK mean?
Response: Changed as suggested (CK is sterile water)
Minor points:
- Be consistent with the abbreviation of X. citri throughout the text
Response: Changed as suggested (X. citri subsp. citri)
- Line 60, give the reference for article describing the genome of X. citri
Response: Changed as suggested
- Fig. 2A, although we can identify the mutant, I suggest labeling the spots to undoubtedly demonstrate what is what; e.g. wild type, delta-phoP, and complemented mutant
Response: Changed as suggested
- Line 370- “avirulence protein”
Response: Changed as suggested
